# Association of Social Vulnerability and COVID-19 Mortality Rates in Texas between 15 March 2020, and 21 July 2022: An Ecological Analysis

**DOI:** 10.3390/ijerph20216985

**Published:** 2023-10-27

**Authors:** Dennis Ogeto Nyachoti, Nalini Ranjit, Ryan Ramphul, Leah D. Whigham, Andrew E. Springer

**Affiliations:** 1Epidemiology and Surveillance Unit, Texas Department of State Health Services, 201 W Howard Ln, Austin, TX 78753, USA; 2Department of Health Promotion and Behavioral Sciences, Center for Community Health Impact, The University of Texas Health Science Center at Houston School of Public Health, 5130 Gateway Boulevard East MCA 110, El Paso, TX 79905, USA; leah.d.whigham@uth.tmc.edu; 3Department of Health Promotion and Behavioral Sciences, The University of Texas Health Science Center at Houston School of Public Health, 1616 Guadalupe, Austin, TX 78701, USA; nalini.ranjit@uth.tmc.edu; 4Department of Epidemiology, Human Genetics & Environmental Sciences, The University of Texas Health Science Center at Houston School of Public Health, 1200 Pressler Street, Houston, TX 77030, USA; ryan.ramphul@uth.tmc.edu

**Keywords:** COVID-19, SARS-CoV-2, Coronavirus 2019, pandemics, SVI, MH SVI, minority health

## Abstract

Background: Despite the key role of social vulnerability such as economic disadvantage in health outcomes, research is limited on the impact of social vulnerabilities on COVID-19-related deaths, especially at the state and county level in the USA. Methods: We conducted a cross-sectional ecologic analysis of COVID-19 mortality by the county-level Minority Health Social Vulnerability Index (MH SVI) and each of its components in Texas. Negative binomial regression (NBR) analyses were used to estimate the association between the composite MH SVI (and its components) and COVID-19 mortality. Results: A 0.1-unit increase in the overall MH SVI (IRR, 1.27; 95% CI, 1.04–1.55; *p* = 0.017) was associated with a 27% increase in the COVID-19 mortality rate. Among the MH SVI component measures, only low socioeconomic status (IRR, 1.55; 95% CI, 1.28–1.89; *p* = 0.001) and higher household composition (e.g., proportion of older population per county) and disability scores (IRR, 1.47; 95% CI, 1.29–1.68; *p* < 0.001) were positively associated with COVID-19 mortality rates. Conclusions: This study provides further evidence of disparities in COVID-19 mortality by social vulnerability and can inform decisions on the allocation of social resources and services as a strategy for reducing COVID-19 mortality rates and similar pandemics in the future.

## 1. Introduction

Coronavirus disease 2019 (COVID-19) remains a significant public health problem globally, including in the United States of America (USA) [1,2,3]. As of 10 March 2023, the USA recorded over 103 million cases and 1 million COVID-19-related deaths while Texas reported above 8.4 million cases and 93,000 COVID-19 deaths [4]. Despite heightened interventions such as COVID-19 vaccination campaigns, the USA and Texas were reporting close to 255 and 17 COVID-19-related deaths daily as of March 2023 [5]. Interestingly, there is a continued emergence of new Omicron subvariants that are more transmissible and can neutralize vaccine antibodies [6,7].

All populations are at risk of developing COVID-19 complications and death; however, people with underlying health conditions and the elderly are more likely to experience worse COVID-19 outcomes [2,8,9]. Several early reports indicated that the COVID-19 pandemic disproportionately impacted racial minority groups and people of low socioeconomic class [10,11,12,13,14]. Later in the pandemic, especially during the COVID-19 third wave (Winter of 2020), after the development of vaccines, whites experienced a higher proportion of deaths [15]. Further research on COVID-19 mortality by county-level social vulnerability holds promise to advance our understanding of the role of Social Determinants of Health (SDoH) and racial/ethnic differences in mitigating or exacerbating the health and quality of life effects of COVID-19 and similar pandemics.

As identifying vulnerable communities is vital to protect populations from the effects of a pandemic [16,17], measures such as the Social Vulnerability Index hold promise to advance our understanding of the effects of social factors such as socioeconomic status on COVID-19-related outcomes. The Social Vulnerability Index (SVI) is a percentile-based measure created by the Centers for Disease Control and Prevention (CDC) to help local officials identify populations at risk of adverse outcomes in the event of a disaster or disease outbreak [18]. Recent evidence has shown that SVI is associated with COVID-19 health and mortality-related outcomes [11,15,19,20,21]. The average death per capita was significantly higher in counties with greater SVI than those with lower SVI scores [15]. Similarly, high composite SVI scores as well as higher scores for each of the individual SVI components were significantly associated with higher mortality rates in USA counties [11].

While this body of research provides an important foundation for understanding the role of social vulnerabilities in relation to COVID-19 health and mortality-related outcomes, these studies used the “traditional” SVI, which is limited to four SVI components only: (1) socioeconomic status, (2) household composition and disability, (3) minority status and language, and (4) housing type and transportation. Furthermore, most of these past studies were conducted during the early months of the pandemic when COVID-19 deaths were still low, thus warranting additional exploration [15,20,21]. Additionally, medical vulnerabilities and healthcare infrastructure access components are among the essential social factors in pandemics and infectious disease outcomes requiring further investigation [17,19,22], yet these components were lacking from the “traditional” SVI.

In addressing limitations of the “traditional” SVI, the Office of Minority Health at the USA Department of Health and Human Services, in partnership with the CDC, recently modified the “traditional” SVI to create the Minority Health Social Vulnerability Index (MH SVI) to help researchers study and plan COVID-19 response programs [23,24]. This adapted index incorporates medical vulnerabilities, healthcare infrastructure access components, and the specific race and language categories that were essentially missing from the “traditional” SVI [23].

In addition to limitations with the use of the “traditional” SVI, research is limited on the impact of social vulnerabilities on COVID-19 mortality at the county level, especially in Texas, which ranks as the sixth most diverse state in terms of minority groups [25] and counts a third of its population (30%) living below 200% of the federal poverty level [26]. In addressing current gaps in the literature and with the aim of advancing our understanding of the role of social vulnerabilities in relation to COVID-19 mortality, this study used an ecological study design to examine the associations between the county-level MH SVI and its individual components and COVID-19 mortality rates in Texas from 15 March 2020 to 21 July 2022. Findings from this study hold promise to inform the design of public health programs, guide the allocation of resources, and formulate policies to reduce COVID-19 mortality rates in Texas and future pandemics.

## 2. Materials and Methods

### 2.1. Study Design and Study Population

This study used a cross-sectional ecological study design suitable for exploring population-level data and comparing mortality rates between counties [27]. We included all counties in Texas (*n* = 254) reporting cumulative COVID-19 deaths among all populations in the analysis. We chose counties as the unit of observation unit as (1) most COVID-19 policies and interventions are approved at the county level [28], (2) local health departments, state, or the CDC report COVID-19 mortality data (for all age populations) aggregated by county, and (3) the MH SVI scores are computed at the county level [23,24].

### 2.2. Study Variables and Data Sources

We used the COVID-19 mortality rate as the outcome of interest. We sourced COVID-19 mortality data between 15 March 2020 and 21 July 2022 from the Texas Department of State and Health Services (TDSHS) website, which are publicly available and de-identified. The TDSHS reports COVID-19-related mortality based on death certificates. Mortality rates (proportions) were calculated as deaths/100,000, i.e., we obtained the ratio of number of deaths (counts) reported per county to the county’s population and multiplied by 100,000 [29]. Table 1 provides details of the study variables and data sources.

We studied the county-level Minority Health Social Vulnerability Index (MH SVI) and its independent components (including their subcomponents) as our exposure variables. The CDC’s Geospatial Research, Analysis & Services Program (GRASP) in partnership with the USA Department of Health and Human Services Office Minority Health (OMH) created the county-level MH SVI using 2014–2018 census bureau data to help public health professionals and emergency responders plan, identify, and support communities in the event of a disaster or disease outbreak such as the COVID-19 response [24]. The index ranks USA counties based on 34 social factors (subcomponents), including unemployment, income levels, racial and ethnic minority status, household challenges (such as percentage of households with no vehicle available), medical health conditions (such as the county’s cardiovascular disease death rate), and access to health services (e.g., number of hospitals per 100,000 population) and further groups them into six related components: (1) socioeconomic status, (2) household composition and disability, (3) minority status and language, (4) housing type and transportation, (5) healthcare infrastructure and access, and (6) medical vulnerability. Each county is then ranked based on the six components and the overall MH SVI score, ranging from 0 to 1, with “0” indicating the least vulnerable and “1” the most vulnerable. The MH SVI calculation and design are documented in the CDC and OMH websites [24,30].

We included the following county-level covariates in the initial analysis based on the literature and a priori knowledge of possible confounding [31]: (1) urbanicity, (2) percentage of adult smokers, (3) population density per square mile, (4) mean life expectancy, (5) average particulate matter (PM)_2.5_, (6) gender, (7) political affiliations (based on 2020 presidential elections), (8) percentage of residents vaccinated against COVID-19 (i.e., received at least one dose of the vaccine), and (9) median age [11,15,21,32]. Table 1 provides details of the covariates and data sources. We did not include the number of primary physicians per 100,000 population in each county, the percentage of persons using public transportation, and the percentage of comorbidities per county [11] because they were incorporated in the MH SVI as specific indicator variables (MH SVI subcomponents).

**Table 1 ijerph-20-06985-t001:** List of Study Variables and Data Sources.

Data	Year	Source
Outcomes		
Mortality	July 2021–July 2022	COVID-19 Cases and Deaths by Vaccination Status Dashboard [33].
Exposures		
Minority Health Social Vulnerability Index (MH SVI)	2018 (Modified July 2021)	Minority Health Social Vulnerability Index [24].
Covariates		
Population Density	2019	County Population Totals: 2010–2019 [34].
Urbanicity	2020	Economic Research Service USA Department of Agriculture (ERS-USDA) [35].
Percentage of Adult Smokers in the County	2019	County Rankings. Smoking [36]
County Mean Life Expectancy	2018–2020	County Rankings. Life expectancy [36].
Percentage of Females per County (Gender)	2018	County Population [34].
Average Particulate Matter (PM)_2.5_	2020	County ranking. Average particulate matter [36]
Percentage of Residents Fully Vaccinated Against COVID-19	2020–2022	COVID-19 Cases and Deaths by Vaccination Status Dashboard [33].
Political Affiliations (2020 Presidential Voting Patterns per County)	2020	MIT Election Data Lab [37].
County’s Median Age	2019	USA Census Bureau [34].

### 2.3. Statistical Analysis

We conducted a descriptive analysis of all study variables for Texas counties and created a bivariate map displaying mortality rates (deaths per 100,000 population) as of 21 July 2022, overlaid on the composite MH SVI for each county in Texas. For the main analyses (i.e., between MH SVI and COVID-19 mortality rates), we used the negative binomial regression (NBR) model because COVID-19 deaths are reported as count data, i.e., only non-negative integers, and it manages overdispersion. To convert COVID-19 death counts to rates, we controlled the population size as the offset variable [29]. Thus, our models examined the relationship between MH SVI (including each individual MH SVI component and its subcomponents) and COVID-19 death counts with a log of population size per county.

We first assessed the relationship between the proposed covariates and COVID-19 mortality, using NBR analysis. Following covariate assessment, we estimated models for the association of each MH SVI component, as well as the composite MH SVI, with mortality rates [10]. Both unadjusted and covariate-adjusted models were estimated, and since MH SVI components and their subcomponents are related to each other, we entered each into a separate NBR model to avoid potential collinearity issues [11]. To facilitate interpretation, we calculated the mortality rate ratio (interpreted as incidence rate ratio, IRR) by exponentiating the regression coefficient [29]. We analyzed data using STATA 17.0 (StataCorp LLC, College Station, TX, USA) and ArcGIS Pro 3.3 (Environmental Systems Research Institute, Inc., Redlands, CA, USA) software and assessed statistical significance at *p* < 0.05. This study was determined exempt by the University of Texas Health Science Center at Houston Committee for the Protection of Human Subjects as it uses publicly available and de-identified secondary data analysis.

## 3. Results

### 3.1. Descriptive Characteristics of Sample

All counties of the state of Texas, USA (*n* = 254 counties), were included in the analysis (Table 2). As of 21 July 2022, COVID-19 mortality per 100,000 population mean was 470.83 in Texas. About twenty percent (19.48%) of the population aged 25+ had no high school diploma, and 16% lived below the federal poverty level. In addition, most minority groups (34.83%) were Hispanic or Latinx, and about 18% of the counties’ population lacked health insurance.

### 3.2. Descriptive Bivariate Map of County-Level MH SVI and COVID-19 Mortality Rates in Texas

Figure 1 exhibits a bivariate choropleth map to visualize the composite MH SVI by COVID-19 mortality per 100,000 population among all populations in Texas, with darker brown colors representing counties with both a greater MH SVI score (most vulnerable, ≥0.75) and the highest COVID-19 mortality rates (≥468 per 100,000 population) mostly in the North, East, and South of Texas. Populous counties such as Dallas, Austin, San Antonio, and El Paso, despite having moderate to high MH SVI vulnerability scores (blue to dark blue colors), showed lower COVID-19 mortality rates.

### 3.3. Associations between the Covariates and COVID-19 Mortality Rates in Texas

In the covariate analyses, we found a significant association between adjusted population density and COVID-19 mortality (IRR, 0.99; 95% CI, 0.99–1.00; *p* < 0.001) (Table 3). Adjusted county mean life expectancy (IRR, 0.95; 95% CI, 0.93–0.96; *p* < 0.001), average particulate matter (IRR, 0.89; 95% CI, 0.86–0.92; *p* < 0.001), percentage of female population (IRR,1.02; 95% CI, 1.01–1.03; *p* = 0.005), and vaccination coverage (receiving at least one dose of COVID-19) (IRR,1.02; 95% CI, 1.02–1.03; *p* < 0.001) as well as political affiliations and urbanicity were associated with COVID-19 mortality rates. Based on these findings, we included all listed covariates in the model.

### 3.4. Associations between MH SVI and Components with COVID-19 Mortality in Texas

The adjusted composite MH SVI was significantly associated with COVID-19 mortality rates (Table 4). A 0.1-unit increase in the overall MH SVI was associated with a 27% increase in mortality rate (IRR, 1.27; 95% CI, 1.04–1.55; *p* = 0.017). Similarly, low socioeconomic status (IRR, 1.55; 95% CI, 1.28–1.89; *p* = 0.001) and greater household composition and disability scores (IRR, 1.47; 95% CI, 1.29–1.68; *p* < 0.001) were associated with an increase in COVID-19 mortality rates in the adjusted model. Adjusted medical vulnerabilities (IRR, 1.14; 95% CI, 0.95–1.38; *p* = 0.157) and housing type and transportation (IRR, 1.06; 95% CI, 0.91–1.23; *p* = 0.425) were not statistically significant at *p* < 0.05. Minority status/language (IRR, 0.94; 95% CI, 0.76–1.16; *p* = 0.572) and healthcare infrastructure and access (IRR, 0.97; 95% CI, 0.83–1.11; *p* = 0.654) were also not associated with COVID-19 mortality rates

### 3.5. Association of MH SVI Subcomponents and COVID-19 Mortality Rates in Texas

We also assessed county-level MH SVI subcomponents and COVID-19 Mortality Rates (Table 5). In the crude model, some of the subcomponents associated with increased COVID-19 mortality rates included the percentage of persons living below poverty levels (IRR, 1.02; 95% CI, 1.00–1.03; *p* < 0.001), unemployment rates (IRR, 1.02; 95% CI, 1.00–1.04; *p* = 0.013), persons aged 25 years and older with no high school diploma (IRR, 1.01; 95% CI, 1.00–1.02; *p* < 0.001), persons aged 65 and older (IRR, 1.02; 95% CI, 1.01–1.03; *p* < 0.001), population with a disability (IRR, 1.03; 95% CI, 1.02–1.05; *p* < 0.001), households with no vehicles (IRR, 1.04; 95% CI, 1.02–1.06; *p* < 0.001), persons without health insurance (IRR, 1.02; 95% CI, 1.01–1.03; *p* < 0.001), persons without internet access (IRR, 1.02; 95% CI, 1.01–1.03; *p* < 0.001), and the county’s cardiovascular disease rate (IRR, 1.01; 95% CI, 1.00–1.01; *p* < 0.001) and respiratory diseases rate (IRR, 1.01; 95% CI, 1.00–1.01; *p* < 0.001). Income per capita (IRR, 0.99; 95% CI 0.99–1.00; *p* < 0.001), percentage of multiple housing units (IRR, 0.96; 95% CI 0.95–0.97; *p* < 0.001), and primary care physicians per 100,000 population (IRR, 0.80; 95% CI 0.68–0.94; *p* = 0.006) were all associated with lower COVID-19 mortality rates.

In the adjusted model, the percentage of persons living below poverty levels (IRR, 1.01; 95% CI, 1.00–1.02; *p* = 0.006), persons aged 25 years and older with no high school diploma (IRR, 1.01; 95% CI, 1.00–1.02; *p* < 0.001), the population aged 65 and older (IRR, 1.04; 95% CI 1.03–1.05; *p* < 0.001), people with a disability (IRR, 1.01; 95% CI 1.00–1.02; *p* = 0.008), households with no vehicles (IRR, 1.03; 95% CI, 1.01–1.05; *p* = 0.002), persons without health insurance (IRR, 1.01; 95% CI, 1.00–1.02; *p* = 0.033), and persons without internet access (IRR, 1.01; 95% CI, 1.01 –1.02; *p* < 0.001) exhibited a persistent positive significant association with COVID-19 mortality rates. The proportion of multiple housing units (IRR, 0.98; 95% CI 0.97–0.98; *p* < 0.001) and income per capita per county (IRR, 0.99; 95% CI 0.99–1.00; *p* < 0.001) were negatively associated with COVID-19 mortality rates.

In the association between minority status (racial/ethnic and language) and COVID-19 mortality rates, a 0.1-unit increase in the percentage of Hispanic or Latinx (IRR, 1.01; 95% CI 1.00–1.01; *p* < 0.001) and Spanish speakers (IRR, 1.01; 95% CI 1.00–1.01; *p* = 0.046) was associated with a 1% increase in COVID-19 mortality rates after controlling for the covariates. Conversely, the percentage of Asian population, Chinese, and Russian speakers were associated with decreased COVID-19 mortality, after adjusting for the covariates.

## 4. Discussion

We examined the association of the composite and components of MH SVI and COVID-19 mortality rates in Texas between 15 March 2020 and 21 July 2022. The adjusted composite MH SVI was significantly associated with increased COVID-19 mortality rates. Several previously published studies using the “traditional” SVI have also reported similar higher mortality rates in counties with greater composite SVI [11,20]. This finding highlights the contribution of the overall vulnerability to increased COVID-19 mortality in Texas counties.

In the individual component analyses, the household composition and disability component was positively associated with COVID-19 mortality rates, which supports previous research [11,15,20]. Further analysis revealed that counties with a higher percentage of a population aged 65 or older and those with a higher proportion of persons with disabilities were also associated with increased COVID-19 mortality rates. These findings underscore COVID-19 mortality disparities experienced among people with disabilities and those counties with a higher percentage of the older population [15,19].

The socioeconomic status component was also positively associated with COVID-19 mortality rates in the crude and adjusted model. Multiple studies have demonstrated similar findings [11,15,21]. Additionally, several socioeconomic status indicators (subcomponents) were also associated with increased COVID-19 mortality rates such as the percentage of persons living below poverty levels, unemployment rates, and individuals with no high school diploma per county. Low socioeconomic status related to COVID-19 including its specific indicators has been reported widely in the literature [16,17].

Although medical vulnerability and healthcare infrastructure and access components were insignificant, several studies have indicated the role of comorbidities and limited access to health in increasing COVID-19 mortality rates [11,21]. However, in the further analysis of the subcomponents, we found a significant increase in COVID-19 mortality rates among counties with a higher proportion of cardiovascular disease deaths and chronic respiratory diseases (unadjusted), individuals without access to the internet (both adjusted and unadjusted), and persons without health insurance (adjusted). These results reaffirm the role of comorbidities, lack of health insurance, and limited access to health information on COVID-19 mortality rates in Texas and across the USA [1,3].

Additionally, while no significant association was found between the minority status and language component with COVID-19, this study found counties with a higher proportion of Hispanic or Latinx population and Spanish speakers were associated with increased COVID-19 mortality rates. This finding concurs with other multiple researchers indicating that Hispanic populations were adversely affected by COVID-19 deaths [12,13,14]. Not all minorities were adversely impacted, however. Inverse associations were seen between the percentage of the Asian population, Chinese, and Russian speakers per county and COVID-19 mortality rates. These findings differ from most past studies showing racial minorities experienced worse COVID-19 outcomes including death [10,11,20]. A possible explanation for these discrepancies would be the recent heightened efforts to mitigate COVID-19 effects such as vaccination efforts and health messaging targeting historically minority groups [38,39]. It is equally likely that some of these groups represent “model minorities”, and may, in fact, have better resources than even the majority white group where they stay. To the authors’ knowledge, this was the first study to examine the association of the specific racial/language minority status within the MH SVI variable and COVID-19 mortality rates. Additionally, association does not imply causality [40]. Thus, further research should be undertaken to explore ethnic-group-specific factors that protect against or exacerbate the risks of mortality from COVID-19.

Specific strengths and limitations of this study merit mention. Among the key strengths, this study is the first to examine the association between the new MH SVI (and its components and subcomponents) and COVID-19 mortality rates in the USA, and is the first, to the authors’ knowledge, to examine the association for the state of Texas. Second, this study adds to the literature by providing a deeper exploration of the specific components and subcomponents of the MH SVI to better understand the existence of various disparities in COVID-19 mortality rates and advance public health interventions aiming to address social environmental indicators of a pandemic. Despite these strengths, we point out specific limitations: First, we did not adjust the COVID-19 mortality rates by age groups as data sourced were aggregated for all populations by county. However, we included each county’s median age as a covariate in the model and the composite MH SVI incorporates the percentage of the population equal to or older than 65 years and younger than 17 years old per county [24]. Second, we cannot rule out the ecological fallacy bias. Thus, findings from this study may not infer to finer levels, such as census tracts, zip codes, or individuals within the counties. With data availability, future studies should focus on smaller observation units. Third, we used datasets collected at different time points and contexts (e.g., most MH SVI and covariates datasets were collected before the pandemic). However, we chose the most recent available data as of the time of the analysis. Fourth, although we included multiple covariates, we acknowledge the impossibility of controlling for all confounders, and thus our study may contain residual confounding. Finally, this study used a cross-sectional ecologic design, and therefore the associations between MH SVI and COVID-19 mortality rates cannot infer causality.

## 5. Conclusions

The adjusted composite MH SVI was significantly and positively associated with COVID-19 mortality rates. Similarly, adjusted socioeconomic status, household composition, and disability components were associated with increased COVID-19 mortality rates between 17 March 2020 and 21 July 2022. Although minority group status and language, housing type and transportation, healthcare infrastructure and access, and medical vulnerability components were insignificant, we found an association between multiple specific subcomponents within the components and COVID-19 mortality rates. These findings underscore the persistence of disparities, especially in the most vulnerable counties, based on the MH SVI scores and the need to explore the specific MH SVI subcomponents (also referred to as SDoH indicators). In addition, this study holds promise to inform decisions related to the allocation of resources aimed at addressing future pandemics similar to COVID-19. Since this was the first study to examine COVID-19 mortality and its association with the MH SVI, future studies should focus on assessing the temporal and spatial association of the MH SVI and COVID-19.

## Figures and Tables

**Figure 1 ijerph-20-06985-f001:**
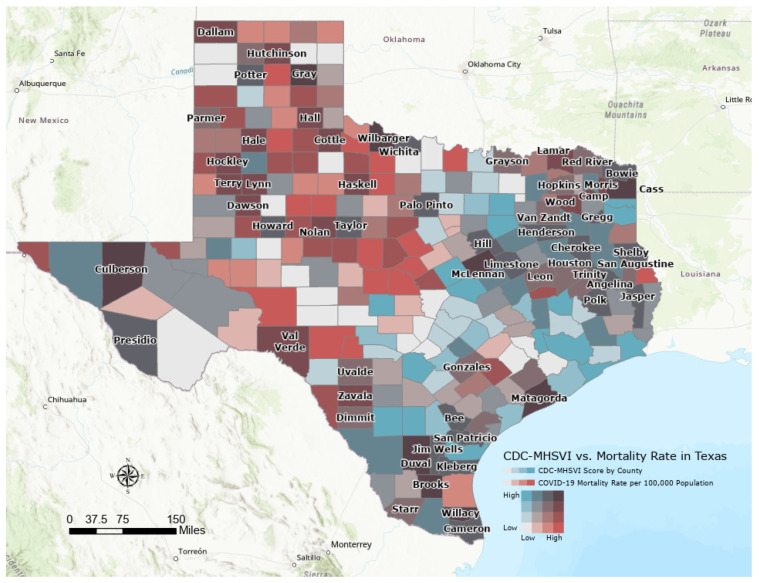
Bivariate map showing the composite county-level MH SVI overlaid on mortality rate per 100,000 population for each county in Texas, March 2020–July 2022.

**Table 2 ijerph-20-06985-t002:** County-level characteristics in Texas between 15 March 2020 and 21 July 2022.

Variable (*n* = 254)	Mean	Standard Deviation (±)	Median	Range
COVID-19 Mortality per 100,000 Population	470.83	170.35	463.14	0.00–1211.31
Composite/Overall MH SVI				
Socioeconomic Status				
% of Persons Living Below Poverty	16.05	6.07	15.20	2.60–39.50
% of Civilians Aged 16+ Unemployed	5.62	2.81	5.40	0.00–17.90
Income Per Capita	25,497.25	5292.63	25,128.5 0	13,350.00–43,439.00
% of Persons Aged 25+ with No High School Diploma	19.48	8.27	17.8	6.30–66.3
Household Composition/ Disability				
% of Persons Aged 65 and Older	17.53	5.31	16.80	8.60–35.9
% of Persons Aged 17 and Younger	24.35	3.96	24.10	12.80–36.60
% of Civilian Noninstitutionalized Population with a Disability	16.08	4.54	15.75	5.3–31.00
% of Single Parent Household with Children Under 18	8.07	3.08	8.5	0–22.70
Minority Status/Language				
% of American Indian/Alaska Native	0.57	0.56	0.40	0.00–3.40
% of Asian	1.09	2.01	0.55	0.00–19.90
% of African American	6.34	6.60	4.15	0.00–33.80
% of Native Hawaiian/Pacific Islander	0.06	0.14	0.00	0.00–1.10
% of Hispanic or Latinx	34.83	23.27	26.55	3.50–99.1
% of Some Other Race	5.77	7.99	3.22	0.00–57.01
% of Spanish Speakers *	9.21	8.33	6.76	0.00–51.36
% of Chinese Speakers *	0.07	0.20	0.00	0.00–1.79
% of Vietnamese Speakers *	0.09	0.21	0.00	0.00–1.32
% of Korean Speakers *	0.03	0.09	0.00	0.00–0.77
% of Russian Speakers *	0.01	0.03	0.00	0.00–0.37
Housing Type/Transportation				
% of Housing Structures with 10 or More Units	3.50	4.61	1.90	0.00–28.10
% of Mobile Homes	15.46	7.92	14.45	1.50–51.40
% of Housing Units with More People than Rooms	4.27	2.69	3.65	0.00–21.20
% of Households with No Vehicle Available	5.05	2.21	5.05	0.00–14.40
% of Persons in Living in Institutionalized Group Quarters	4.25	6.17	1.60	0.00–36.50
Healthcare Infrastructure and Access				
Hospitals per 100,000 Persons	6.68	10.7	3.49	0.00–82.55
Urgent Care Clinics per 100,000 Persons	0.61	2.27	0.00	0.00–27.52
Pharmacies per 100,000 Persons	18.55	11.83	17.53	0.00–72.20
Primary Care Physicians per 100,000 Persons	0.38	0.28	0.33	0.00–1.39
% of Persons Without Health Insurance	17.38	4.60	17.10	4.70–33.40
Medical Vulnerability				
Total Cardiovascular Disease Death Rate per 100,000	496.56	97.89	485.80	314.70–834.80
Chronic Respiratory Diseases Rate per 100,000	63.86	17.56	63.64	26.91–113.39
Total Persons with Obesity **	31,087.01	5547.05	30,300.00	20,500.00–47,700.00
Total Persons Diagnosed with Diabetes **	10,042.91	4318.40	9400.00	2300.00–26,800.00
% of persons with No Internet access	25.13	8.37	24.95	6.54–55.02
Covariates				
Population Density	44.81	132.39	8.36	0.06–1145.19
Political Affiliations				
Trump Voters *n* =232, 91.34%	-	-	-	-
Biden Voters *n* = 22, 8.66%	-	-	-	-
% Female	49.06	3.10	50.15	29.37–53.27
Urbanicity				
Suburban *n* = 82, 32.28%	-	-	-	-
Suburban *n* = 84, 33.07%	-	-	-	-
Rural *n* = 88, 34.65%	-	-	-	-
% of Adult Smokers in the County	14.94	1.56	14.89	10.64–19.87
% of Residents Received At Least One Dose of the COVID-19 Vaccine	53.16	14.07	50.85	0.62–78.00
County Mean Life Expectancy	77.46	2.33	77.45	71.86–89.65
Average Particulate Matter (PM)_2.5_	8.68	1.34	8.75	5.60–12.00
County Median Age	39.56	5.74	39.15	27.90–60.00

* Those who speak English less than “very well”; ** age-adjusted (adults aged 20+ years); % = percentage; - = not applicable as the variable is categorical; data sources/dates (described in Table 1): COVID-mortality data (as of 21 July 2022); MH SVI (2014–2018 Census data).

**Table 3 ijerph-20-06985-t003:** Association between Covariates and COVID-19 Mortality Rates, Texas, 15 March 2020–21 July 2022.

	Unadjusted COVID-19 Mortality Rate ^a,c^			Adjusted COVID-19 Mortality Rate ^b,c^		
	IRR	95% CI	*p*-Value	IRR	95% CI	*p*-Value
Covariates ^d^ (*n* = 254)						
Population Density	0.99	0.99–1.00	**<0.001**	0.99	0.99–1.00	**0.001**
Political Affiliations						
Trump Voters	REF			REF		
Biden Voters	0.89	0.76–1.04	0.136	1.27	1.10–1.45	**0.001**
% Female	1.01	0.99–1.02	0.763	1.02	1.01–1.03	**0.005**
Urbanicity						
Urban	REF			REF		
Suburban	1.35	1.22–1.48	**<0.001**	1.12	1.03–1.21	**0.005**
Rural	1.36	1.23–1.51	**<0.001**	1.03	0.94–1.14	0.483
Percentage of Adult Smokers in the County	1.07	1.04–1.10	**<0.001**	1.04	1.01–1.06	**0.005**
Vaccination Coverage (receiving at least one dose of the COVID-19 vaccine)	0.99	0.98–0.99	**<0.001**	0.99	0.99–1.00	**0.013**
County Mean Life Expectancy	0.94	0.92–0.95	**<0.001**	0.95	0.93–0.96	**<0.001**
Average Particulate Matter (PM)_2.5_	0.89	0.86–0.92	**<0.001**	0.89	0.86–0.92	**<0.001**
County Median Age	1.01	1.00–1.02	**0.028**	1.01	1.00–1.01	**0.018**

Abbreviations: IRR = incidence rate ratio; CI = confidence interval; REF = reference. Symbols: *p* < 0.05; % = percent. ^a^ Crude model (unadjusted). ^b^ Adjusted model (controlled for each other). ^c^ Mortality rates were estimated using negative binomial regression (NBR). ^d^ Each variable was entered into a separate NBR model to assess association with COVID-19 mortality [11]. Notes: All regression models included an offset for the total number of people residing in the county. **Bold**: Statistically significant at *p* < 0.05.

**Table 4 ijerph-20-06985-t004:** Association between the composite county-level MH SVI, its components, and COVID-19 Mortality Rates, Texas, 15 March 2020–21 July 2022.

Variable ^d^	Unadjusted COVID-19 Mortality Rate ^a,c^			Adjusted COVID-19 Mortality Rate ^b,c^		
	IRR	95% CI	*p*-Value	IRR	95% CI	*p*-Value
Composite/Overall MH SVI	1.10	0.91–1.33	0.328	1.27	1.04–1.55	**0.017**
Socioeconomic Status	1.95	1.62–2.35	**<0.001**	1.55	1.28–1.89	**0.001**
Household Composition/ Disability	1.86	1.62–2.15	**<0.001**	1.47	1.29–1.68	**<0.001**
Minority Status/Language	0.52	0.42–0.63	**<0.001**	0.94	0.76–1.16	0.572
Housing Type/Transportation	0.95	0.81–1.12	0.554	1.06	0.91–1.23	0.425
Healthcare Infrastructure and Access	1.23	1.01–1.51	0.038	0.97	0.83–1.11	0.654
Medical Vulnerabilities	1.72	1.44–2.06	**<0.001**	1.14	0.95–1.38	0.157

Abbreviations: IRR = incidence rate ratio; CI = confidence interval. Symbols: *p* < 0.05; % = percent; ^a^ Crude model (unadjusted). ^b^ Adjusted model (controlled for population density, vaccination coverage, estimated percentage of female population, mean life expectancy, average particulate matter per county, percentage of adult smokers in the county, urbanicity, political affiliations, and median age). ^c^ Mortality rates were estimated using negative binomial regression (NBR). ^d^ Each variable was entered into a separate NBR model to assess association with COVID-19 mortality [11]. Notes: All regression models included an offset for the total number of people residing in the county. **Bold**: Statistically significant at *p* < 0.05.

**Table 5 ijerph-20-06985-t005:** Association between the county-level MH SVI subcomponents and COVID-19 Mortality Rates, Texas, 15 March 2020–21 July 2022.

Variable ^d^	Unadjusted COVID-19 Mortality Rate ^a,c^			Adjusted COVID-19 Mortality Rate ^b,c^		
	IRR	95% CI	*p*-Value	IRR	95% CI	*p*-Value
Socioeconomic Status						
% of Persons Living Below Poverty	1.02	1.00–1.03	**<0.001**	1.01	1.00–1.02	**0.006**
% of Civilians Aged 16+ Unemployed	1.02	1.00–1.04	**0.013**	1.01	1.00–1.03	0.093
Income Per Capita	0.99	0.99–1.00	**<0.001**	0.99	0.99–1.00	**<0.001**
% of Persons Aged 25+ with No High School Diploma	1.01	1.01–1.02	**<0.001**	1.01	1.00–1.02	**<0.001**
Household Composition/ Disability						
% of Persons Aged 65 and Older	1.02	1.01–1.03	**<0.001**	1.04	1.03–1.05	**<0.001**
% of Persons Aged 17 and Younger	1.01	0.99–1.01	0.940	1.01	0.99–1.02	0.459
% of Civilian Noninstitutionalized Population with a Disability	1.03	1.03–1.05	**<0.001**	1.01	1.00–1.02	**0.008**
% of Single Parent Household with Children Under 18	1.01	1.00–1.03	0.064	1.02	1.01–1.03	**0.005**
Minority Status/Language						
% of American Indian/Alaska Native	1.06	0.97–1.15	0.206	0.98	0.92–1.04	0.592
% of Asian	0.92	0.90–0.93	**<0.001**	0.97	0.95–0.98	**<0.001**
% of African American	0.99	0.98–1.00	**0.003**	0.99	0.98–1.01	0.096
% of Native Hawaiian/Pacific Islander	0.90	0.66–1.21	0.489	0.97	0.78–1.20	0.816
% of Hispanic or Latinx	1.01	0.99–1.00	0.138	1.01	1.00–1.01	**<0.001**
% of Some Other Race	0.99	0.99–1.00	0.585	0.99	0.99–1.00	0.601
% of Spanish Speakers *	1.01	1.00–1.01	0.554	1.01	1.00–1.01	**0.046**
% of Chinese Speakers *	0.56	0.47–0.69	**<0.001**	0.76	0.65–0.89	**0.001**
% of Vietnamese Speakers *	0.58	0.46–0.72	**<0.001**	0.87	0.73–1.03	0.114
% of Korean Speakers *	0.42	0.27–0.66	**<0.001**	0.79	0.55–1.12	0.185
% of Russian Speakers *	0.05	0.01–0.18	**<0.001**	0.19	0.07–0.51	**0.001**
Housing Type/Transportation						
% of Housing Structures with 10 or More Units	0.96	0.95–0.97	**<0.001**	0.98	0.97–0.98	**<0.001**
% of Mobile Homes	1.01	0.99–1.01	0.802	0.99	1.00–1.00	0.509
% of Housing Units with More People than Rooms	1.01	0.98–1.02	0.747	1.01	0.99–1.03	0.120
% of Households with No Vehicle Available	1.04	1.02–1.06	**<0.001**	1.03	1.01–1.05	**0.002**
% of Persons in Living in Institutionalized Group Quarters	0.99	0.99–1.00	0.479	0.99	0.99–1.01	0.428
Healthcare Infrastructure and Access						
Hospitals per 100,000 Persons	1.01	1.00–1.01	**0.014**	0.99	0.99–1.00	0.571
Urgent Care Clinics per 100,000 Persons	0.99	0.98–1.01	0.590	1.01	0.99–1.02	0.582
Pharmacies per 100,000 Persons	1.01	1.00–1.01	**0.015**	1.01	1.00–1.01	0.224
Primary Care Physicians per 100,000 Persons	0.80	0.68–0.94	**0.006**	1.01	0.88–1.13	0.978
% of Persons Without Health Insurance	1.02	1.01–1.03	**<0.001**	1.01	1.00–1.02	**0.033**
Medical Vulnerability						
Total Cardiovascular Disease Death Rate per 100,000	1.01	1.00–1.01	**<0.001**	1.00	0.99–1.01	0.970
Chronic Respiratory Diseases Rate per 100,000	1.01	1.00–1.01	**<0.001**	0.99	0.99–1.00	**0.027**
Total Persons with Obesity **	1.01	0.99–1.01	0.725	1.00	0.99–1.00	0.947
Total Persons Diagnosed with Diabetes **	1.01	0.99–1.00	0.620	1.00	0.99–1.01	0.653
% of Persons with No Internet Access	1.02	1.02–1.03	**<0.001**	1.01	1.01–1.02	**<0.001**

Abbreviations: IRR = incidence rate ratio; CI = confidence interval; *p* < 0.05; % = percent; ^a^ Crude model (unadjusted). ^b^ Adjusted model (controlled for population density, vaccination coverage, estimated percentage of female population, mean life expectancy, average particulate matter per county, percentage of adult smokers in the county, urbanicity, political affiliations, and median age). ^c^ Mortality rates were estimated using negative binomial regression (NBR). ^d^ Each variable was entered into a separate NBR model to assess association with COVID-19 mortality [11]. Notes: All regression models included an offset for the total number of people residing in the county. **Bold**: Statistically significant at *p* < 0.05; * Those who speak English less than “very well”. ** Adults aged 20+; Age-adjusted.

## Data Availability

The datasets supporting the conclusions of this article are available de-identified and publicly available. The outcome (COVID-19 Mortality) data are available from the Texas Department of State Health Services website: https://www.dshs.texas.gov/covid-19-coronavirus-disease-2019/covid-19-vaccine-information/covid-19-cases-deaths (accessed on 25 January 2023) and the dependent variable (Minority Health Social Vulnerability Index) is available from the Health and Human Services Minority Health website: https://www.minorityhealth.hhs.gov/minority-mental-health (accessed on 25 January 2023).

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
