# Peer review of "Association of Social Vulnerability and COVID-19 Mortality Rates in Texas between 15 March 2020, and 21 July 2022: An Ecological Analysis"

_ijerph, 2023, doi:10.3390/ijerph20216985_

Round 1

Reviewer 1 Report

Comments and Suggestions for Authors

Thank you for the opportunity to review the manuscript entitled “Association of Social Vulnerability and COVID-19 Mortality Rates in Texas between March 15, 2020, and July 21, 2022: An Ecological Analysis” (IJERPH-2610258). The paper aims to study the association between the detailed components of social vulnerability index (SVI) and COVID-19 mortality in Texas. The topic is interesting and important, and overall well-written. But some parts of the methods and results are confusing and could be improved. Below I raise up my suggestions in no particular order.

1.     Since the major aim of the study is to examine the detailed components of SVI, the measures of these components and sub-components, and how they are combined together should be discussed concretely in the method section (section 2.2 on page 3). Now in the second paragraph of section 2.2 (line 108-120) the authors only listed the names of these components and how the components are used to calculate SVI, but no information on subcomponents and how these subcomponents are used to compute the measure of the 6 components.

Another minor here: on page 3 line 114, the number of medical vulnerability should be “6)”. Now there are two “5)”. 

2.     On page 4 section 2.3 line 144-145, the description is incorrect: Table 2 is about descriptive statistics of all variables, not the results of NBR analysis. 

3.     For Tables 2-5, why “Minority Status/Language” and “Language” are two separate sections? It is confusing since previously in section 2.2 they are in the same category. I suggest only including one or combining them somehow. 

4.     I have some questions about section 2.3 and sections 3.3-3.5. 

1)    First, section 2.3 should contain more details about the analytical approach, including the mapping (shown in Figure 1, which I like a lot), and the details about analysis shown in Tables 3-5. Especially the footnotes “d” under Tables 3-5 should be discussed in the section 2.3. For Table 3.4, I understand the overall MH SVI cannot be included in the model together with its components, but why the components cannot be included together? The same for Table 3.5, why the variables were entered into NBR model separately? Is there other research using the same approach that you can cite? For Table 3.5, why can’t variables of a specific component be entered into the model together, and why can’t all variables entered into the model together? 

2)    For the title line of Tables 3-5, you can just put “unadjusted” and “adjusted” in the name of the columns, and explain what “adjusted” and “unadjusted” mean in the footnote. It’s confusing to use the same wording for two columns (both “COVID-19 Mortality Rate”) and add the difference in one footnote. 

3)    The components you examined throughout the analysis are not consistent. “Medical Vulnerability” is missing in Table 2. “Housing Type/Transportation” is missing in Table 5. Please check carefully.

Minor:

Table 2, on page 5 line 164, the line “% of Housing Units with More People than Rooms” is bold, but should not be. 

Comments on the Quality of English Language

Minor: Some sentences are framed awkwardly. For example on page 2 line 80-82, the sentence “research is limited on the impact of social vulnerabilities on COVID-19 mortality at the county level, which includes states such as Texas, which is ranked as the sixth most diverse state in terms of minority groups” includes two “which.”

Reviewer 2 Report

Comments and Suggestions for Authors

In this manuscript, Ogeto Nyachoti et al. examined the association of the composite and components of MH SVI and COVID-19 mortality rates in Texas and found the adjusted composite MH SVI was significantly associated with increased COVID-19 mortality rates. The paper is well-written, and the results are supported by the data. I only have minor comments:

- The number or percentage of COVID-19 cases in Texas should be indicated.

Comments on the Quality of English Language

The English language seems fine to me.

Reviewer 3 Report

Comments and Suggestions for Authors

The Authors of the study investigated the impact of social vulnerability on COVID-19-related deaths in Texas at the county level by using the Minority Health Social Vulnerability Index (MH SVI) and its components. Overall, the manuscript addresses a pertinent research question presents valuable findings, and has the potential to be published.

However, I am afraid that the study has not undergone any Institutional Review Board. Hence, it is essential to ensure that the study followed ethical guidelines and obtained appropriate approvals before conducting the research. If it is not applicable kindly state the reason for the same.

Reviewer 4 Report

Comments and Suggestions for Authors

I would like to express my sincere thanks to the authors for the opportunity to review this exciting manuscript.

I have only one small, albeit relevant, comment:

The tables are difficult to access due to the same header content distinguished only by the superscript numbers. The comprehensibility of the table content should be improved at this point by an appropriate adjustment.

Reviewer 5 Report

Comments and Suggestions for Authors

Dear authors:

congratulation to your article.

Aim:

Authors conducted a cross-sectional  ecologic analysis of COVID-19 mortality by county-level Minority Health Social Vulnerability Index 21 (MH SVI) and each of its components in Texas.

exact definition : these studies used the “traditional” SVI, which is limited to four SVI components  only:

1) socioeconomic status,

2) household composition and disability,

3) minority status 66 and language,

4) housing type and transportation.

tabs: p. 5

....

- well structured article

- personal involment of the paper

- interesting and important article

Interesting results:  "Conversely, the percentage of Asian population, Chinese, and Russian speakers were associated with decreased COVID-19 mortality, after adjusting for the covariates." line 235 !

Problem: please add 2-3 paragraphs, correct your CONCLUSION. - (compare quality of the parts of your article (methods, introduction, tabs, with your Conclusion )

please add to your Introduction ideas about well-being, happiness, values connection with COVID

Murgaš, F., Petrovič, F., Maturkanič, P., & Kralik, R. (2022). Happiness or Quality of Life? Or Both?. Journal of Education Culture and Society, 13(1), 17–36. https://doi.org/10.15503/jecs2022.1.17.36

Maturkanič, P.; Čergeťová, I.T.; Králik, R.; Hlad, Ľ.; Roubalová, M.; Martin, J.G.; Judák, V.; Akimjak, A.; Petrikovičová, L. The Phenomenon of Social and Pastoral Service in Eastern Slovakia and Northwestern Czech Republic during the COVID-19 Pandemic: Comparison of Two Selected Units of Former Czechoslovakia in the Context of the Perspective of Positive Solutions. Int. J. Environ. Res. Public Health 2022, 19, 2480. https://doi.org/10.3390/ijerph19042480

Kralik, R. (2023). The Influence of Family and School in Shaping the Values of Children and Young People in the Theory of Free Time and Pedagogy. Journal of Education Culture and Society, 14(1), 249–268. https://doi.org/10.15503/jecs2023.1.249.268

I recommend publishing your article.

Round 2

Reviewer 1 Report

Comments and Suggestions for Authors

Thanks for the response and editing. The MS looks good to me.